# Perceptual Discrimination of Phonemic Contrasts in Quebec French: Exposure to Quebec French Does Not Improve Perception in Hexagonal French Native Speakers Living in Quebec

Scott Kunkel [1,*], Elisa Passoni [1] and Esther de Leeuw [2]

1   Department of Linguistics, Queen Mary University of London, London E1 4NS, UK
2   Department of English, University of Lausanne, CH-1015 Lausanne, Switzerland
*   Correspondence: s.kunkel@qmul.ac.uk

**Abstract:** In Quebec French, /a ~ ɑ/ and /ɛ ~ aɛ/ are phonemic, whereas in Hexagonal French, these vowels are merged to /a/ and /ɛ/, respectively. We tested the effects of extended exposure to Quebec French (QF) as a second dialect (D2) on Hexagonal French (HF) speakers' abilities to perceive these contrasts. Three groups of listeners were recruited: (1) non-mobile HF speakers born and living in France (HF group); (2) non-mobile QF speakers born and living in Quebec (QF group); and mobile HF speakers having moved from France to Quebec (HF>QF group). To determine any fine-grained effects of second dialect (D2) exposure on the perception of vowel contrasts, participants completed a same–different discrimination task in which they listened to stimuli paired at different levels of acoustic similarity. As expected, QF listeners showed a significant advantage over the HF group in discriminating between /a ~ ɑ/ and /ɛ ~ aɛ/ pairs, thus suggesting an own-dialect advantage in perceptual discrimination. Interestingly, this own-dialect advantage appeared to be greater for the /ɛ ~ aɛ/ contrast. The QF listeners also showed an advantage over the HF>QF group, and, surprisingly, this advantage was greater than over the HF group. In other words, the results suggested that the acquisition of a second dialect did not enhance the abilities of listeners to perceive differences between phonemic contrasts in that D2. If anything, the acquisition of the D2 disadvantaged the perceptual abilities of the HF>QF group. This might be because these phonemes have, over time, become less acoustically marked for the HF>QF participants and have, potentially, become integrated into their D1 phonemic categories.

**Keywords:** speech perception; perceptual adaptation; vowel discrimination; vowel contrasts; second dialect acquisition; Quebec French

## 1. Introduction

Work focusing on speech production has shown that speakers can acquire non-native vowel contrasts in their speech as a consequence of extended exposure to a second dialect (D2) of their native language (Chambers 1992) and that this can occur even after moving to a new D2 region in adulthood (Johnson 2007; Nycz 2013; Walker 2019). In comparison, relatively few studies have explored whether speakers' perception of non-native vowel contrasts is also capable of changing as a result of D2 exposure. However, there is some evidence that long-term exposure to a D2 can result in an improved ability to perceive phonemic vowel categories in the D2 that are not native in a listener's first dialect (D1) (Bowie 2000); although, other studies have failed to find similar evidence of perceptual change in adulthood among other D2 populations (Ziliak 2012). Therefore, the extent to which D1 perceptual categories are susceptible to change after long-term D2 exposure in adulthood is not yet well understood.

The current study seeks to shed light on D1 perceptual malleability by examining how extended exposure to a D2 of French can influence the perception of non-native vowel

contrasts. Specifically, we examine two vocalic contrasts in spoken Quebec French (QF): /a ~ ɑ/ (1) and /ɛ ~ aɛ/ (2).

1. *tache* [taʃ] 'stain' vs. *tâche* [tɑʃ] 'task, chore'
2. *mettre* [mɛtʁ] 'to put' vs. *maître* [maɛtʁ] 'owner, teacher'

While these vowels are lexically contrastive in QF—as shown by the minimal pairs in (1) and (2)—these contrasts have largely been neutralized in Hexagonal French (HF, i.e., the Standard French of France) with only [a] and [ɛ] being maintained in the majority of HF speakers' phonemic inventories (Hansen and Juillard 2011). Given this situation of cross-dialectal variation, exploring how HF speakers who have moved to Quebec in adulthood perceive these contrasts—and whether they perceive the contrasts at all—can contribute to a better understanding of the extent to which native speech perception is malleable in post-adolescence.

## 2. Background

### 2.1. Cross-Dialectal Perception of Mergers

Research on the cross-dialectal perception of mergers has typically shown listeners who do not have a phonemic contrast in their D1 to be worse at perceiving this contrast in speech, as compared to speakers who do have this contrast natively (e.g., Conrey et al. 2005; Janson and Schulman 1983; Labov et al. 1991). This has often been shown through lexical discrimination tasks, in which participants hear two words and are asked to report whether the words they hear are the same or different from one another. Practically speaking, however, this method is known to be difficult to calibrate. For instance, several researchers have run into the issue of how to interpret results of participants who perform well above chance in accurately discriminating between words containing non-native regional contrasts (e.g., Austen 2020; Bowie 2000; Labov et al. 1991; Thomas and Hay 2005). This has led on occasion to listeners who do not show 100% accuracy in distinguishing minimal pairs to be considered for analytic purposes as perceptually merged. However, as Wade (2017) points out, this high cut-off point does not preclude the possibility of listeners still picking up on relevant perceptual cues between such word pairs.

One method that has been used to avoid such ceiling effects involves having listeners discriminate words that are acoustically resynthesized along a gradient continuum between two phonemic categories. For instance, Fridland and Kendall (2012) used a seven-step resynthesized continuum between vowels [e] to [ɛ] in US English to test at what point American English speakers change their perception from one vowel to the other when played ambiguous tokens of word pairs *bait* [beɪt]–*bet* [bɛt] and *date* [deɪt]–*debt* [dɛt]. They found that participants from the Southern US, a variety characterized by the merging of these vowels to [e], perceived more of the vowel steps along this continuum as [e] compared to non-Southern, non-merged participants. This was evidenced by a later crossover point of the gradient stimuli (i.e., when perception of [e] or [ɛ] is at 50%) for the Southern group versus the non-Southern group. This and other studies examining the perception of gradient phoneme contrasts (e.g., Bukmaier et al. 2014; Miller et al. 2011) have shown that listeners attend to fine-grained and dialect-specific information in an utterance when mapping an incoming speech signal onto existing vowel categories.

### 2.2. Perceptual Adaptation in the Acquisition of a New Language Variety

Research on second dialect acquisition (SDA) has typically focused on changes in production that come with long-term contact with a D2. Past SDA studies having explored perceptual change have often found individuals with more exposure to a D2 to hold advantages in perceiving D2 speech, as compared to speakers of the same D1 with less exposure to this D2 (e.g., Scott and Cutler 1984; Bowie 2000; Evans and Iverson 2004, 2007; Walker 2018; Voeten 2021). For instance, Walker (2018) showed that mobile English speakers who moved from the US to the UK, and vice versa, were unsurprisingly better at transcribing speech in noise spoken in their non-native dialect (American or British English) as compared to non-mobile counterpart speakers of these dialects. Moreover,

other studies have shown listeners with D2 experience to be better at perceiving specific phonemic distinctions present in a D2 that are absent in their D1. For instance, Scott and Cutler (1984) showed that, compared to British speakers who had never lived outside of Britain, British speakers with extensive experience living in the US were faster at resolving lexical ambiguities between words involving [t] and [ɾ], a distinction present in American but not British English. More recently, Bowie (2000) found evidence that exposure to a dialect containing a vowel contrast which is absent in another dialect of the same language can result in an improved ability to discriminate between these contrastive, non-native vowels in perception. This study showed that listeners who lived outside Waldorf were generally better than listeners who never lived outside Waldorf at discriminating between pre-lateral /u/, /ʊ/, and /o/, vowel categories that are merged in the Waldorf dialect (i.e., the listeners' D1).

However, strong evidence of perceptual adaptation has not always been found among listeners with long-term exposure to a D2. For example, Ziliak (2012) tested the perceptual categorization of vowels among non-mobile listeners from Southern Indiana and Chicago and mobile speakers who had moved from Indiana to Chicago. Participants heard words resynthesized along continua designed to test the perception three features of the Northern Cities Shift (/a/-fronting, /æ/-raising, and /ɛ/-backing). In a two-alternative forced choice design, participants selected between one of two words to indicate the word they heard. For all three continua tested, no statistically significant differences in categorical perception were found between listeners who had moved from Southern Indiana to Chicago and non-mobile residents of Southern Indiana. Although some intra-group variability was found among the mobile participants in terms of how closely they patterned in their perception of the features as compared to life-long Chicago residents, this was also the case for the non-mobile Indiana participants.

Studies of perceptual attrition in contexts of SLA also shed light on the extent to which perceptual adaptation can occur in adolescence. Like exposure to a D2, linguistic competition from a second language (L2) can result in perceptual changes to the native speech perception system. For example, Celata and Cancila (2010) tested the perception of singleton and geminate consonant distinctions (e.g., *casa* [kasa] 'house' vs. *cassa* [kasːa] 'box') in the Italian Lucchese dialect among three groups of participants: (i) first-generation speakers born in Lucca who had moved to San Francisco in adulthood, (ii) second-generation Lucchese speakers living in the US and born to immigrant parents from Lucca, and (iii) non-mobile speakers born and living in Lucca. It was found that the first-generation Lucchese speakers who had moved to the US in adulthood were better at discriminating between words with the singleton–geminate contrast than the second-generation speakers who were born in the US. In addition, they reported that both the first- and second-generation immigrant groups in the US showed a disadvantage in accurate identification of words containing singleton consonants versus geminates compared to the non-mobile Lucchese speakers. The higher error rate of the first-generation participants compared to the non-mobile participants suggests that L2 acquisition in adulthood can result in changes to the native L1 perceptual system and indeed hinder L1 perception.

Considered together, several observations can be drawn from this past research concerning the effects of late D2 or L2 acquisition on perceptual malleability in the native linguistic system. The results of Scott and Cutler (1984), Bowie (2000), and Walker (2018) indicate that individuals' speech perception abilities can change in adulthood after extended exposure to a D2. Moreover, these studies show that perceptual adaptation can take place in a number of domains, including the fine-grained perception of specific phonemes (e.g., Bowie 2000) and the comprehension of speech more generally (e.g., Walker 2018). Celata and Cancila (2010) and other recent studies of perceptual restructuring in late bilingualism (e.g., Cabrelli et al. 2019; de Leeuw et al. 2021; Tobin et al. 2017) lend additional support to the view that perceptual adaptation is possible in post-adolescence.

However, as shown by the results of Ziliak (2012), perceptual change (like changes in production, see Nycz 2015) is not guaranteed to occur as a result of speaker mobility.

Therefore, more research needs to be conducted to understand the mechanics of post-adolescent perceptual malleability among mobile speakers. The present study focuses on perceptual adaptation as relating to vowel discrimination, a dimension of perceptual malleability that remains underexplored. This is undertaken by exploring whether mobile listeners who do not have a native vowel contrast in their D1 perceptually discriminate tokens containing this contrast differently from listeners who do not have the same amount of exposure to this D2. Furthermore, while it has been shown that listeners' ability to perceptually discriminate between two words that differ as a function of a vowel contrast depends in part on the vocalic inventory of their native language (Amengual and Chamorro 2015) or dialect (Riverin-Coutlée and Arnaud 2015), it remains unclear whether the ability to discriminate between non-native vowel contrasts in perception can be acquired after extended exposure to a D2. The present study seeks to shed light on these questions through examining the perception of two vocalic contrasts in QF (/a ~ ɑ/ and /ɛ ~ aɛ/). Would the HF>QF speakers outpreform the HF group, or would they subceed?

*2.3. Variables of Interest*

Although having roots in the history of HF, /a/ and /ɑ/ (e.g., *tache* [taʃ] 'stain' vs. *tâche* [tɑʃ] 'task, chore') have been undergoing a merger in France whereby the back vowel /ɑ/ is realized close to or identical to the front vowel /a/ (Berns 2019). This merger has resulted in the phonemic neutralization of possible minimal pairs no longer commonly contrastive in HF, such as *patte* 'paw' ~ *pâte* 'dough' and *tache* 'stain' ~ *tâche* 'task'. Despite the fact that most speakers in France show a merger in the pronunciation of /a/ and /ɑ/ (ibid.), there is still some amount of inter-speaker variation found in the pronunciation of /a/ and /ɑ/ (Hansen and Juillard 2011), particularly in Eastern France along the Belgian and Swiss borders (Avanzi 2017). On the other hand, the /a ~ ɑ/ vowel contrast is actively present in QF and is indeed considered one of the most characteristic markers of this dialect (Brasseur 2009). Because of the ubiquity of this feature, as well as the fact that it is not subject to negative social evaluation, this low contrast is often noted to constitute part of the norm of pronunciation for QF (e.g., Chalier 2021). This is supported by the fact that it is used widely even in formal speech contexts such as Radio-Canada news broadcaster speech (Bigot and Papen 2013). Thus, the marked difference in the presence of /a ~ ɑ/ in QF vs. HF makes it an ideal variable for studying the perception of cross-dialectal phonetic variation.

The second contrast considered in this study is the mid-vowel contrast /ɛ ~ aɛ/ (e.g., *mettre* [mɛtʁ] 'to put' vs. *maître* [maɛtʁ] 'owner, teacher'). One important difference between the /a ~ ɑ/ contrast and the /ɛ ~ aɛ/ contrast lies in the respective historical origins of these contrasts. While /a ~ ɑ/ has deep roots in the history of HF (Berns 2015), the diphthongization of long vowels in French, as in /ɛ ~ aɛ/, are 'phonetic creations' of QF (Reinke and Ostiguy 2016, p. 55), having originated after French colonial contact with North America and therefore possessing no historical counterpart in HF[1]. Like the /a ~ ɑ/ contrast, the contrast in /ɛ ~ aɛ/ is ubiquitous in contemporary QF (e.g., Côté and Lancien 2019; Riverin-Coutlée and Roy 2020). In QF, the long vowel [ɛː] shows a strong tendency to diphthongize to [aɛ], with recent acoustic studies showing this phoneme to be very often or even categorically realized with a complex nucleus (e.g., Leblanc 2012; Riverin-Coutlée and Roy 2020). Moreover, this contrast, *unlike* the /a ~ ɑ/ contrast, is subject to social stigmatization and is not typically considered part of the QF norm of pronunciation (Chalier 2021, p. 313).

In summary, these two contrasts were considered of interest in this study due to their cross-dialectal differences in HF and QF: (1) /a ~ ɑ/ represents a contrast which is being merged, i.e., 'lost' in HF while maintained in QF; (2) /ɛ ~ aɛ/ represents a contrast which is not present in HF and is therefore a phonetic creation in QF. The difference in social evaluation of these contrasts in QF also made them interesting variables to compare. We were curious to explore whether these differences would be noticeable in our results as, essentially, the mobile participants would have potentially encountered /ɑ/ in France, but /aɛ/ would be an entirely new phoneme to them in their D2.

*2.4. Research Question and Hypotheses*

As discussed in more detail below, three groups of listeners (i.e., (1) non-mobile HF speakers born and living in France (HF group), (2) non-mobile QF speakers born and living in Quebec (QF group), and (3) mobile HF speakers having moved from France to Quebec (HF>QF group) were recruited to investigate the following research question:

- What effect does extended exposure to QF as a D2 have on HF>QF's ability to discriminate between these contrasts?

We predicted to find an effect of mobility on HF>QF listeners' ability to discriminate between /a ~ ɑ/ and /ɛ ~ aɛ/, such that the HF>QF participants would prove *better* at accurately discriminating between acoustically similar items containing these contrasts as compared to the HF participants, given the HF>QF group's increased exposure to these contrasts in QF as an ambient D2. If supported, this finding would suggest that perceptual adaptation had taken place in the mobile speakers' native linguistic systems. It was further predicted that the HF>QF listeners would perform at an intermediary level between the HF and QF listeners, with the QF listeners holding an advantage and thus showing more accurate discrimination of these contrasts overall. The methods for testing this hypothesis are discussed in more detail in the following section.

## 3. Materials and Methods

*3.1. Stimuli Design*

Stimuli were recorded by a native female speaker of QF, who was 22 years old and who had spent her entire life in Quebec (in the Bois-Francs region) at the time of recording. Recordings were made in a quiet room at Université de Sherbrooke using an Olympus LS-7 Linear PCM Recorder audio recorder at a mono, 32-bit sampling rate of 44.1 kHz. The speaker was asked to read two word lists two times each in their most regular and natural accent. The list comprised words with the target vowel contrasts (e.g., *tache-tâche*, *mettre-maître*) as well as words with other phonemic contrasts across HF and QF (/u/ vs. /y/, e.g., *bout–but*, /ɛ̃/ vs. /œ̃/, e.g., *brin–brin*).

Eight minimal pairs were selected for this task. All sixteen words show medium to relatively low lexical frequency; specifically, they appeared less than 300 times but more than 5 times per million words in the *Lexique 2* corpus of written French (New et al. 2004) (however, note that two lower-frequency words were also used as fillers; see Appendix A). Where possible, attempts were made to match the grammatical category of the words in each pair; unfortunately, this was not always possible due to the small number of possible QF minimal pairs containing the contrasts examined (e.g., *laide* 'ugly' ~ *l'aide* 'the help', *mettre* 'to put' ~ *maître* 'master').

As they are most relevant to the research question at hand, only the four target word pairs (i.e., those containing /a ~ ɑ/ and /ɛ ~ aɛ/—Table 1) will be discussed in the remainder of this paper.

**Table 1.** Cross-dialectal comparison of the eight word pairs (four target pairs and four distractor pairs) included in the same–different task.

| Phonemic Contrast | Minimal Pairs | Pronunciation in HF | Pronunciation in QF |
|:---:|:---:|:---:|:---:|
| /a ~ ɑ/ | patte 'paw' ~ pâte 'dough'<br>tache 'stain' ~ tâche 'task' | [pat] ~ [pat]<br>[taʃ] ~ [taʃ] | [pat] ~ [pɑt]<br>[taʃ] ~ [tɑʃ] |
| /ɛ ~ aɛ/ | laide 'ugly' ~ l'aide 'the help'<br>mettre 'to put' ~ maître 'master' | [lɛd] ~ [lɛd]<br>[mɛtʁ] ~ [mɛtʁ] | [lɛd] ~ [laɛd]<br>[mɛtʁ] ~ [maɛtʁ] |
| /u ~ y/ | boule 'ball' ~ bulle 'bubble'<br>bouche 'mouth' ~ bûche 'log' | [bul] ~ [byl]<br>[buʃ] ~ byʃ] | [bʊl] ~ [bʏl]<br>[bʊʃ] ~ [bʏʃ] |
| /o ~ ɔ/ | saute 'jump' ~ sotte 'dumb' | [sot] ~ [sɔt] | [sot] ~ [sɔt] |
| /e ~ ɛ/ | épée 'sword' ~ épais 'thick' | [epe] ~ [epɛ] | [epe] ~ [epɛ] |

3.1.1. Resynthesis

In the present study, resynthesized vowel continua were used to test whether individuals who had moved from France to Quebec, where they had received extended exposure to QF as a D2, perceptually differentiate and categorize the QF contrasts /a ~ ɑ/ and /ɛ ~ aɛ/. In Praat (Boersma and Weenink 2021), gradient vowel continua for each of the eight minimal pairs in Table 1 were created using a script written by Lawrence (2018) and previously used in similar work (e.g., Alderton 2020; Barnard 2021). This script uses LPC (linear prediction coding) to estimate the spectral envelope of a speech sound. An iterative inverse-filtering technique (based on Alku et al. 1999) is then used to render a voicing source representation, allowing users to create an acoustic gradient continuum between two specified, natural speech tokens. Once the resynthesized segments are modified and their pitch and amplitude contours are matched to those of the vowels in the natural tokens, they are embedded back into the original consonantal context, retaining the original flanking environment of one of the natural tokens. This approach was chosen because it (1) embeds high-end components of the speech signal back into the modified segment to produce less synthetic-sounding stimuli and (2) allows users to manipulate both spectral quality and segment duration in the resynthesis process. The latter was deemed particularly important for this study given the role that duration plays in the production of these two vowels in QF (Riverin-Coutlée and Roy 2020).

Vowel trajectory and duration were not manipulated independently from one another in the resynthesis process for two reasons. First, vowel duration has been shown to be closely intertwined with vowel diphthongization in QF, such that vowel lengthening is a necessary condition for diphthongization to occur in QF (Riverin-Coutlée and Roy 2020). It was therefore reasoned that separating these two cues would not reflect the phonological reality of this variable in spoken QF. Second, as previously stated, recent acoustic studies have shown the monophthongal realization of words such as *maître* 'master' as [mɛːtʁ] to be largely marginal (Leblanc 2012) or even absent (Riverin-Coutlée and Roy 2020) in modern-day QF.

This process yielded eight unique nine-step vowel continua. Four of these continua contained the target contrasts /a ~ ɑ/ and /ɛ ~ aɛ/ and the other four contained other contrasts used as distractors. Midpoint F1/F2 formant values for the four resynthesized target continua are plotted in the right panel of Figure 1 (for more detailed formant measurements for each of these steps, see Appendix B). To ensure that these resynthesized tokens could be generalized as representative of QF, their spectral properties were compared to those of Riverin-Coutlée and Roy (2020), who conducted an acoustic analysis of 37 female QF speakers of similar age to the speaker in this study (ranging from 18 to 23 years). As shown below in Figure 1, the resynthesized tokens fall within a realistic spectral range when compared to the naturalistic vowel spaces of native, female QF speakers. The endpoints of each continuum (i.e., steps 1 and 9) approximately correspond in acoustic space to the formant values of the natural (i.e., unsynthesized) stimuli recorded by the speaker, while still being synthesized along with the other continua steps. Comparing the two target contrasts, one finds that each of the two minimal pairs containing the /ɛ ~ aɛ/ contrast overlap to a great degree in their respective vowel spaces, while there is no spectral overlapping of the continuum steps for the pairs involving the /a ~ ɑ/ contrast (with the *patte-pâte* continuum being consistently higher in the vowel space than the *tache-tâche* continuum).

Figures 2 and 3 show the formant trajectories of every other step in the *tache-tâche* and *mettre-maître* continua, respectively (more detailed measures for the resynthesized steps of each continuum are included in Appendix B).

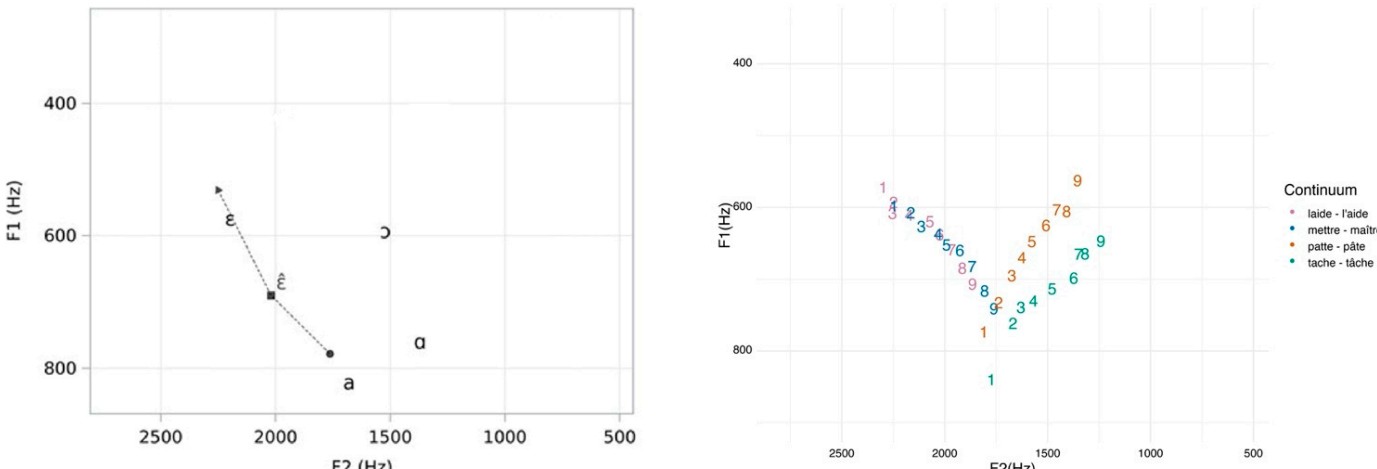

**Figure 1.** (**Left**) Vowel chart adapted from Riverin-Coutlée and Roy (2020, p. 237) showing F1/F2 midpoint values for 37 female native Quebec speakers. The complex /aɛ/ vowel is represented at three timesteps: 25% (shown by the circle), 50% (shown by the square), and 75% (shown by the arrow). (**Right**) F1/F2 midpoint values for resynthesized vowel continuum testing the perception of contrasts /a ~ ɑ/ (i.e., *patte-pâte*, *tache-tâche*) and /ɛ ~ aɛ/ (i.e., *laide-l'aide*, *mettre-maître*). The number on the righthand plot represents F1/F2 midpoint values for each of the nine steps of each continuum.

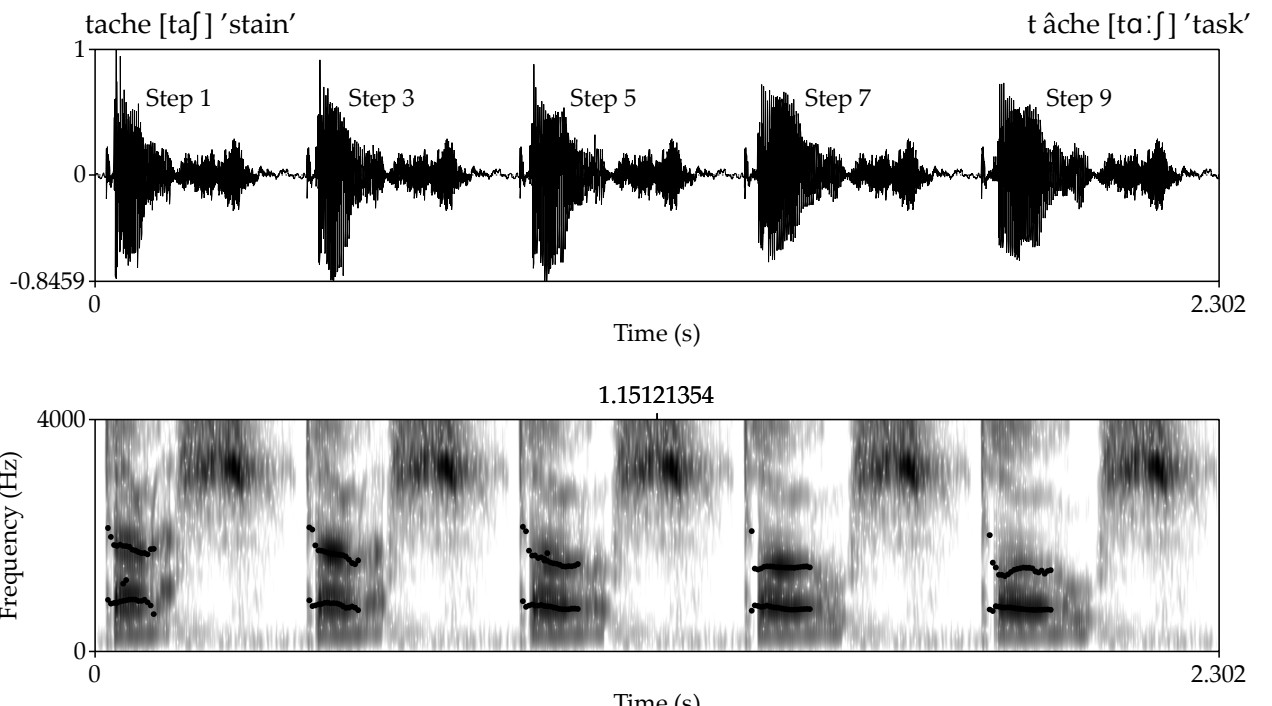

**Figure 2.** Waveform, spectrogram, and F1-F2 trajectories for steps 1, 3, 5, 7, and 9 of resynthesized continuum between /a/ and /ɑ/ in the QF minimal pair *tache-tâche*. The lowering of the F2 formants across the continuum corresponds to an increasingly backed realization of /a/ over the nine steps.

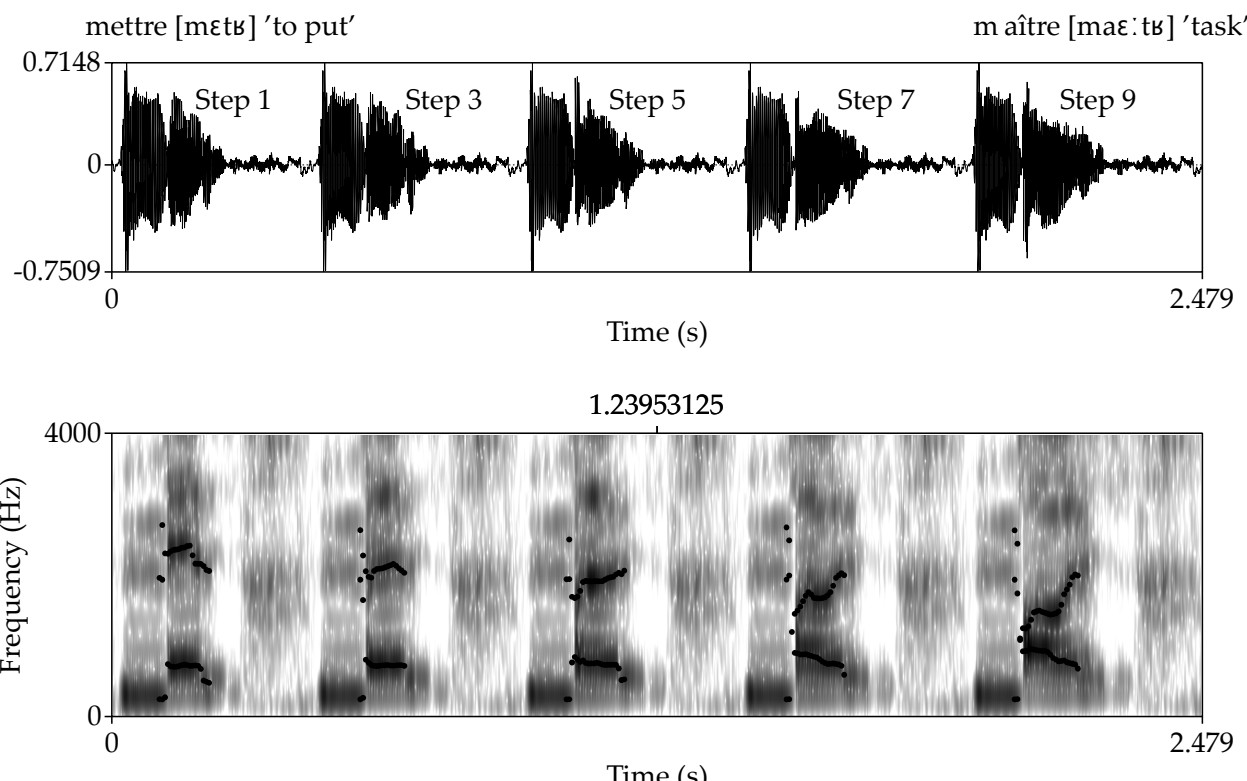

**Figure 3.** Waveform, spectrogram, and F1-F2 trajectories for steps 1, 3, 5, 7, and 9 of resynthesized continuum between [ɛ] and [aɛ] contrast in the QF minimal pair *mettre-maître*. The spectral proximity of the onset of the F1-F2 formants over the course of the continuum indicates progressively more diphthongization of [ɛ].

### 3.1.2. Naturalness Rating Experiment

An online naturalness rating experiment was conducted to test the perceived naturalness of each of the nine steps of each of resynthesized continua. This was conducted to ensure that, as far as possible, the stimuli were assessed on the basis of the extent of the phonemic contrast rather than on the basis of sounding 'unnatural' or 'unnatural'. Ten native French-speaking participants from France, otherwise unrelated to this study, were recruited from the online participant recruitment platform Prolific (www.prolific.co, accessed on 20 October 2022). Participants were presented 72 tokens (i.e., all tokens) on a scale from 1 to 10, with 1 indicating that the token sounded *tout à fait naturel* 'completely natural' and 10 indicating that the tokens sounded *clairement synthétisé* 'clearly synthesized'. Each stimulus was presented once in a randomized order and participants had no time limit to give their response. Before the main experiment, participants were played a recording of the word *joue* ('play') that would merit a rating of 1 (i.e., a natural, unmodified recording of a word) and stimuli that would merit a rating of 10 (i.e., a token manipulated beyond clear lexical recognition). On average, it took 4 min and 19 s (min = 3:21, max = 7:22) to complete the experiment. Average ratings are shown in Table 2; as can be seen, the target continuum received numerically similar naturalness ratings, with the lowest rating (indicating a higher level of perceived naturalness) being 4.96 for *tache-tâche* and the highest rating (indicating a lower level of perceived naturalness) being 6.03 for *laide-l'aide*.

**Table 2.** Mean naturalness ratings for each step of resynthesized target continua (1 = completely natural and 10 = clearly synthesized).

| Continuum | Mean (SD) | 1 | 2 | 3 | 4 | 5 | 6 | 7 | 8 | 9 |
|---|---|---|---|---|---|---|---|---|---|---|
| *patte-pâte* | 5.25 (2.5) | 5.3 | 4.2 | 4.5 | 4.9 | 5 | 5.7 | 6 | 5.7 | 6 |
| *tache-tâche* | 4.96 (2.6) | 2.5 | 3.7 | 4.2 | 4.2 | 6.2 | 5.1 | 6.7 | 6.6 | 5.5 |
| *l'aide-laide* | 6.03 (2.7) | 4.9 | 4.1 | 5.3 | 4.4 | 5.6 | 8 | 6.3 | 7.5 | 8.2 |
| *mettre-maître* | 5.17 (2.9) | 2 | 3.7 | 3.6 | 2.6 | 5.2 | 6.9 | 7.1 | 7.7 | 7.8 |

Table 2 also shows that resynthesized tokens towards the end of each continuum were perceived as less natural on average than the earlier steps in the continuum; this is particularly true for the /ɛ ~ aɛ/ continua (e.g., *l'aide-laide* and *mettre-maître*) and is most likely a reflection of the relative complexity of the spectral properties of these tokens (i.e., the fact that there is more diphthongization over the course of the vowel) compared to the /a ~ ɑ/ continua.

In sum, while the resynthesis process used to create the stimuli in this study did result in tokens that were perceived overall as more synthetic- than natural-sounding, this was, for the purposes of this study, deemed a necessary trade-off for being able to closely control the spectral progression of the stimuli tested. We briefly discuss this trade-off in more detail in the discussion section.

*3.2. Participants*

Thirty-five individuals participated in this study. Participants were recruited into one of three groups depending on their nationality and mobility status:

1. HF group: Non-mobile HF speakers born, raised, and living in France (*n* = 13).
2. HF>QF group: Mobile HF speakers born in France but living in Quebec after moving there in adulthood (*n* = 12).
3. QF group: Non-mobile QF speakers born, raised, and living in Quebec (*n* = 10).

More information about participants is given in Table 3.

**Table 3.** Group data for participant groups.

| Participant Group | Gender Ratio (F:M) | Age (Mean (SD)) | Mean Age of Migration to Quebec | Mean Length of Residence in Quebec |
|---|---|---|---|---|
| HF group (*n* = 13) | 5:8 | 42.8 (8) | n/a | n/a |
| QF group (*n* = 10) | 4:6 | 34.4 (10) | n/a | n/a |
| HF-QF group (*n* = 12) | 9:3 | 45.7 (7.5) | 22 | 7 |

Participants in the HF and QF groups were recruited through Prolific (www.prolific.co, accessed 28 February 2022). Participants in the HF group were between 30 and 58 years of age (mean = 42.8 years; SD = 8 years), whereas participants in the QF group were between 23 and 54 years of age (mean = 34.4 years; SD = 10 years). Participants in the HF and QF groups reported that they spoke French as a native language, had not lived outside their native region (France or Quebec, respectively) for more than one year, and lived in their native region at the time of participation.

The HF>QF participants were recruited through Facebook groups for French expatriates living in Quebec (e.g., *Les Français de la ville de Québec*). They were between 29 and 67 years of age (mean = 45.7 years, SD = 7.5 years) when they moved to Quebec and had lived there between 3 and 42 years (mean = 13.6 years, SD = 9.9 years) by the time of testing. All HF>QF participants reported being born in France, having lived there until they were eighteen, and speaking French as a native language. Table 4 reports detailed information about each of the HF>QF participants.

**Table 4.** HF>QF participants (*n* = 12) having moved from France to Quebec (LoR = length of residence).

| Gender | Age | Age of Migration (in Years) | LoR in Quebec (in Years) | Residence in Quebec | Birthplace in France |
|---|---|---|---|---|---|
| F | 30 | 27 | 3 | Gatineau | Firminy |
| F | 33 | 28 | 5 | Quebec City | Nogent-sur-Marne (Paris) |
| F | 29 | 22 | 7 | Gatineau | Cambrai |
| F | 35 | 27 | 8 | Trois-Rivières | Rennes |
| M | 47 | 37 | 10 | Quebec City | Villerupt |
| F | 49 | 38 | 11 | Portneuf | Marseille |
| F | 63 | 50 | 13 | Quebec City | Saint-Étienne |
| M | 50 | 33 | 17 | Quebec City | Toulon |
| M | 41 | 23 | 18 | Mont-Saint-Hilaire | Sarcelles (Paris) |
| F | 53 | 35 | 18 | Gatineau | Annonay |
| F | 51 | 29 | 22 | Quebec City | Conflans Ste Honorine (Paris) |
| F | 67 | 25 | 42 | Gatineau | Paris |

*3.3. Procedure*

The study was designed and administered online through Gorilla Experiment Builder (www.gorilla.sc, accessed 23 February 2022; Anwyl-Irvine et al. 2020). Participants first verified that they fulfilled the specific participation criteria for their group (as detailed in Section 2.2). They were then asked to read a study description and sign a consent form before progressing to the main portion of the study. Although both production and perception data were collected from participants[2], only one of the perceptual tasks (a same–different discrimination task) is discussed in this paper. This task served to examine participants' ability to differentiate between the target vowel contrasts in perception. The whole study took around 30 min in total to complete, with the discrimination task taking 4–7 min to complete. Participants were asked to wear headphones and complete the study in a quiet space. In all cases, the production data were elicited after participants completed the perception tasks to avoid perceptual priming effects (cf. de Leeuw et al. 2021). At the end of the experimental portion of the study, participants filled in a questionnaire about their language background, language attitudes towards QF, and, in the case of the HF-QF participants, their perceived dialect change since moving to Quebec. Participants were paid either CAD 15 or EUR 10.

For the same–different discrimination task, participants heard two successive words (e.g., *patte-pâte*) and were asked to report whether the pronunciations of the two words were the same (*même*) or different (*différent*) (Figure 4). To investigate the extent to which participants could discriminate varying degrees of acoustic difference between two words, different steps on each continuum were paired in eleven configurations (see, e.g., Amengual and Chamorro 2015; Pallier et al. 1997). For the target trials, that is, those comprising 'different' stimuli, the continua steps were paired in four different configurations with progressively increasing levels of perceptual similarity[3]:

- Steps 1 and 9
- Steps 2 and 8
- Steps 3 and 7
- Steps 4 and 6

Investigating the extent to which participants can perceive small spectral differences in stimuli involving Quebec-specific vowel contrasts allowed us to test the hypothesis that the HF>QF listeners would perform at an intermediary level between the HF and QF groups (suggesting that restructuring has occurred in their native linguistic systems).

Participants were instructed to press the F key on their keyboard with their left finger if they thought that the pronunciations of the two words were identical and to press the J key with their right finger if they thought that the pronunciations of the two words were different. The task was designed so that participant could only respond after hearing the two words in full. Participants were asked to respond as quickly and as accurately as possible after the end of the second word in the pair. To encourage spontaneous responses,

a timeout of 2000 ms from the end of the second stimulus was set before the task advanced automatically to the next trial (cf. de Leeuw et al. 2021).

An inter stimulus interval of 500 ms of silence separated the first word from the second in each trial. Fifteen additional 'same' trials consisting of tokens from the non-target contrast continua (i.e., /u ~ y/, /o ~ ɔ/, /e ~ ɛ/) were also included to even out the numbers of 'same' vs. 'different' trials. In total, participants listened to 60 trials, with 32 'different' pairings and 28 'same' pairings. For each participant, stimuli were presented in a different pseudo-randomized order, such that trials containing tokens along the same lexical continuum (e.g., pairings 1–9 and 3–7 of the *patte-pâte* continuum) did not occur in succession. Participants were given the option to take two breaks at evenly spaced intervals across the task. As it has been shown that listeners' success in discriminating vowel sounds can depend partly on the order in which the stimuli are presented (Francis and Ciocca 2003), each 'different' pair in this task was played twice with the presentation order of the stimuli switched each time.

To ensure participants understood the instructions, they were asked to complete four initial practice trials, after which they were given feedback on the accuracy of their responses. Practice trials comprised two 'same' (1–1 pairings) and two 'different' trials (1–9 pairings) of the *bouche-bûche* continuum, which was not used elsewhere in this task.

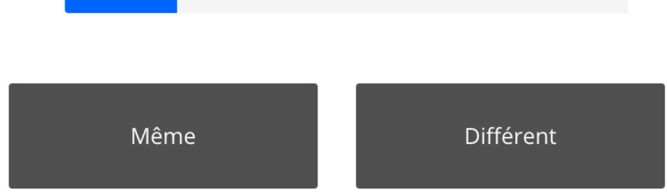

**Figure 4.** Screenshot of online procedure in the same–different discrimination task, as seen by participants. The blue line shows progression of the task.

*3.4. Statistical Analysis*

A total of 2100 responses were collected from this task. Fifteen trials were lost because participants timed out in their response and responses from one QF participant (*n* = 60) were excluded from the analyses as they performed below chance (d' score[4] = −2.16) (the full dataset, Dataset S1: datasetLanguages.csv, can be accessed at https://drive.google.com/drive/folders/1fLlyHjgYJXcMdq8t2HcwaJM-PMPyu6kP?usp=sharing (accessed 5 July 2023)) Table 5 reports mean accuracy for 'same' and 'different' trials and relative d' scores for each group of participants. In general, participants showed similar overall sensitivity, as confirmed by the numerically similar d' scores across groups. With regard to the 'same' trial, participants performed similarly across groups, whereas the same cannot be said for the 'different' trials. As a consequence, and in line with previous similar work (e.g., Amengual and Chamorro 2015), the analysis reported below was only on the 'different' trials (*n* = 1077).

The statistical analysis was carried out in R (R Core Team 2023; version 4.3.0) using the packages *lme4* (Bates et al. 2015) and *lmerTest* (Kuznetsova et al. 2017). All plots were created using *ggplot2* (Wickham et al. 2016). We performed a series of generalized linear mixed effects models (GLMERs) for correct responses for each configuration separately (e.g., step 1–9; step 2–8; step 3–7; step 4–6) to ensure that the potential emergence of significant findings was not an artefact of mere acoustic differences between the two paired stimuli, which could have been, in principle, detected by any French-speaking listeners. Any potential effect of the different configurations was then extrapolated from the analysis by assessing whether factors behaved (dis)similarly across steps (see the comparative sociolinguistics methods, Tagliamonte (2013), for a similar approach).

**Table 5.** Mean accuracy and d' scores for 'same' and 'different' trials in same–different task by participant group. Numbers in the 'Score' columns represent number of correct responses out of total trials in each category.

| | HF Group | | HF>QF Group | | QF Group | |
|---|---|---|---|---|---|---|
| | **Score** | **Rate (%)** | **Score** | **Rate (%)** | **Score** | **Rate (%)** |
| 'Same' | 339/363 | 93.4 | 316/332 | 95.2 | 233/252 | 92.5 |
| 'Different' | 287/415 | 68.9% | 225/377 | 59.7% | 234/285 | 82.1% |
| d' | 1.82 | | 2.05 | | 2.00 | |

In all models, the dependent variable was the proportion of correctly identified 'different' trials. *Group* (HF, QF, HF>QF), *Contrast* (/ɛ ~ aɛ/ vs. /a ~ ɑ/), and their interaction were entered as fixed effects, with *Participant* included as a random intercept (model syntax = DV ~ group * contrast + (1|participant)). Models including random intercepts per word as well as by-participant and by-word random slopes were initially tested but failed to converge. The significance of each models' coefficient was estimated using the Satterthwaite method from *lmerTest* (Kuznetsova et al. 2017) and significance level was set at $p < 0.05$. Following previous work (e.g., Winter and Grawunder 2012; Passoni et al. 2022), if results revealed significant interactions between predictors, any potential main effect was not expanded upon as main effects are uninterpretable in case of significant interactions. Residual plots were visually inspected to detect any obvious deviation from normality and homoscedasticity. Post hoc analyses were run using the *emmeans* package (Lenth 2023) with levels of significance Bonferroni-adjusted for pairwise comparisons.

**4. Results**

Figures 5 and 6 below report percentages of correct answers (i.e., when the participant chose 'different') divided by contrasts and configuration (see Appendix C for percentages of correct answers divided by contrasts and configuration for each of the HF>QF participants). Again, to answer our research question, a series of generalized linear mixed effects models (GLMERs) for correct responses for each configuration separately (see Section 3.4) were run.

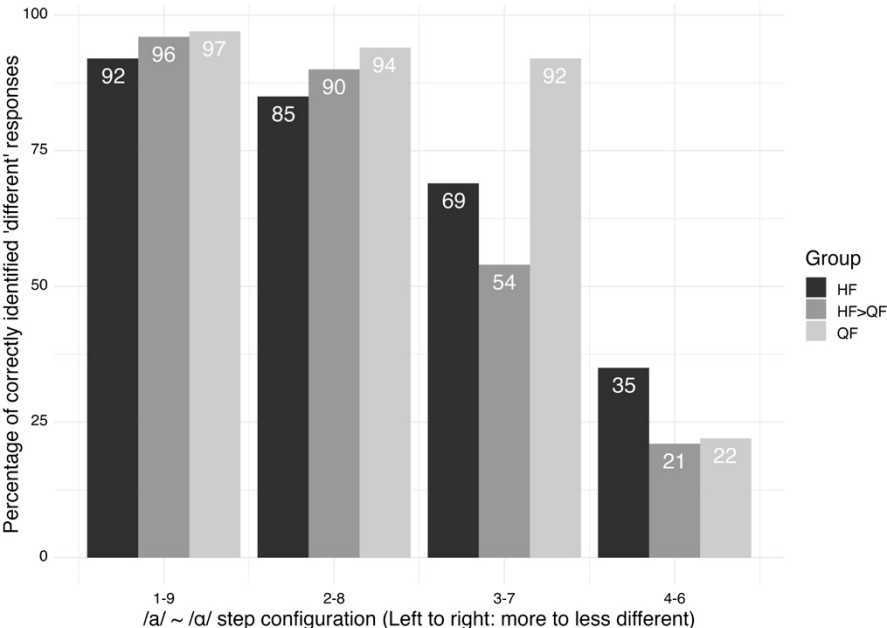

**Figure 5.** Bar plot showing percentage of correctly perceived 'different' discriminations of /a ~ ɑ/ step configurations (*n* = 539). The labels on each of the bars show the percentage of correct discrimination for each participant group for each 'different' step configuration (i.e., 1–9, 2–8, 3–7, 4–6).

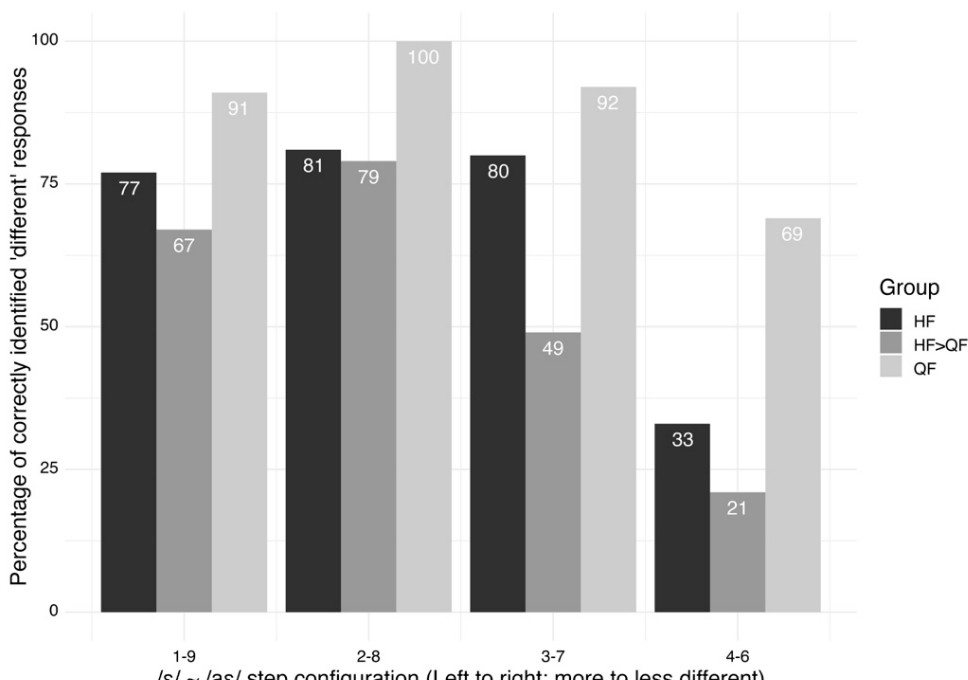

**Figure 6.** Bar plot showing percentage of correctly perceived 'different' discriminations of /ɛ ~ aɛ/ step configurations (*n* = 538). The labels on each of the bars show the percentage of correct identification for each participant group for each 'different' step configuration (i.e., 1–9, 2–8, 3–7, 4–6).

### 4.1. Configuration 1–9

Table 6 below displays the model summary for the configuration 1–9, i.e., the configuration which contained the stimuli which were most different from one another. The main effect of contrast (mid, /ɛ ~ aɛ/ vs. low, /a ~ ɑ/) suggested that, overall, participants were more likely to respond correctly (i.e., choose 'different') when listening to the low vowel contrast than the mid vowel contrast (*p* = 0.034). However, as evident in Figures 5 and 6, there were no significant group differences as, unsurprisingly, all groups performed well on this configuration and approached ceiling for /a ~ ɑ/ (low contrast). For the mid contrast /ɛ ~ aɛ/, the QF group approached ceiling with HF>QF performing least accurately and HF performing between QF and HF>QF; although, again, these group differences were not significant.

**Table 6.** Estimates, standard errors, *z*-values, and *p*-values for the *glmer* model for configuration 1–9.

|  | Estimate | Std. Error | *z*-Value | *p*-Value |
|---|---|---|---|---|
| (Intercept = group = HF, contrast = low) | −2.59 | 0.56 | −4.62 | <0.0001 |
| group = HF>QF | −0.64 | 0.91 | −0.71 | 0.479 |
| group = QF | −1.08 | 1.16 | −0.93 | 0.35 |
| contrast = mid | 1.31 | 0.62 | 2.11 | 0.034 |
| group = HF>QF, contrast = mid | 1.15 | 1.01 | 1.14 | 0.252 |
| group = QF, contrast = mid | −0.11 | 1.33 | −0.08 | 0.929 |

## 4.2. Configuration 2–8

Due to multicollinearity issues, the original model was split by contrast and the *p*-value was therefore Bonferroni-adjusted to 0.025. In addition, because the two new models both revealed singularity issues, for this configuration, we ran two *glm* models with *Group* as fixed effect (syntax of final model$_{2-8}$ = *glm*(DV ~ group)). As can be noted in Tables 7 and 8, neither model detected an effect of group on the dependent variable. Furthermore, as can be observed in Figures 5 and 6, and similarly to configuration 1–9, all groups performed closer to ceiling on the low-contrast /a ~ ɑ/, whereas, for the mid-contrast /ɛ ~ aɛ/, it was the QF group that performed most accurately; although, again, these descriptive differences were not significant.

**Table 7.** Estimates, standard errors, *z*-values, and *p*-values for the *glm* model for configuration 2–8, for the /a ~ ɑ/ contrast.

|  | Estimate | Std. Error | *z*-Value | *p*-Value |
|---|---|---|---|---|
| (Intercept = group = HF) | −1.7 | 0.38 | −4.43 | <0.0001 |
| group = HF>QF | −0.44 | 0.60 | −0.73 | 0.463 |
| group = QF | −1.09 | 0.82 | −1.33 | 0.182 |

**Table 8.** Estimates, standard errors, *z*-values, and *p*-values for the *glm* model for configuration 2–8, for the /ɛ ~ aɛ/ contrast.

|  | Estimate | Std. Error | *z*-Value | *p*-Value |
|---|---|---|---|---|
| (Intercept = group = HF) | −1.44 | 0.35 | −4.08 | <0.0001 |
| group = HF>QF | 0.10 | 0.50 | 0.20 | 0.841 |
| group = QF | −18.13 | 1792.34 | −0.01 | 0.992 |

## 4.3. Configuration 3–7

As displayed in Table 9 for configuration 3–7, the main effect of group (HF vs. HF>QF vs. QF) revealed that, for both contrasts, QF participants were significantly more likely to respond correctly (i.e., choose 'different') than HF participants (ß = 1.83, *z*-value = 2.17, *p* = 0.03) when hearing these paired trials, indicating, very generally, that as the stimuli resembled each other more, the QF participants displayed an own-dialect advantage over the other groups.

**Table 9.** Estimates, standard errors, *z*-values, and *p*-values for the *glmer* model for configuration 3–7.

|  | Estimate | Std. Error | *z*-Value | *p*-Value |
|---|---|---|---|---|
| (Intercept = group = HF, contrast = low) | −0.94 | 0.42 | −2.22 | 0.027 |
| group = HF>QF | 0.74 | 0.61 | 1.22 | 0.222 |
| group = QF | −1.83 | 0.84 | −2.17 | 0.030 |
| contrast = mid | −0.69 | 0.49 | −1.40 | 0.163 |
| group = HF>QF, contrast = mid | 0.96 | 0.68 | 1.41 | 0.160 |
| group = QF, contrast = mid | 0.69 | 1.01 | 0.68 | 0.497 |

As observed in Figures 5 and 6, HF>QF participants performed least accurately on this configuration: on the mid /ɛ ~ aɛ/ contrast, they performed below chance, and on the low-contrast /a ~ ɑ/, they performed just slightly above chance. To further explore the

significant effect of group on the responses for this configuration, we reran the model with QF as baseline. As can be noted in Table 10, the model output revealed that QF participants were also significantly more accurate (i.e., choose 'different') than HF>QF participants (ß = 2.75, *z*-value = 2.99, *p* = 0.002). Notably, as observed in Figures 5 and 6, the HF>QF participants performed less accurately than the HF participants, although this difference was not significant.

**Table 10.** Estimates, standard errors, *z*-values, and *p*-values for the *glmer* model for configuration 3–7 with QF as baseline.

| | Estimate | Std. Error | *z*-Value | *p*-Value |
|---|---|---|---|---|
| (Intercept = group = QF, contrast = low) | −2.77 | 0.73 | −3.74 | 0.0001 |
| group = HF | 1.83 | 0.84 | 2.17 | 0.030 |
| group = HF>QF | 2.57 | 0.86 | 2.99 | 0.002 |
| contrast = mid | 0.79 | 0.82 | 0.00 | 0.999 |
| group = HF contrast = mid | −0.68 | 1 | −0.67 | 0.497 |
| group = HF>QF, contrast = mid | 2.72 | 1 | 0.27 | 0.785 |

*4.4. Configuration 4–6*

Table 11 below shows the model summary for the 4–6 configuration which contained the stimuli which were most similar to one another. The significant interaction between group and contrast was further explored with post hoc analyses (Table 12).

**Table 11.** Estimates, standard errors, *z*-values, and *p*-values for the *glmer* model for configuration 4–6.

| | Estimate | Std. Error | *z*-Value | *p*-Value |
|---|---|---|---|---|
| (Intercept = group = HF, contrast = low) | 0.70 | 0.35 | 1.99 | 0.047 |
| group = HF>QF | 0.70 | 0.53 | 1.30 | 0.192 |
| group = QF | 0.63 | 0.58 | 1.10 | 0.272 |
| contrast = mid | 0.09 | 0.43 | 0.22 | 0.827 |
| group = HF>QF, contrast = mid | −0.06 | 0.67 | −0.10 | 0.924 |
| group = QF, contrast = mid | −2.26 | 0.71 | −3.17 | 0.002 |

As evident in Figure 5, there were no significant group differences for /a ~ ɑ/ (low contrast), as all groups performed below chance on this configuration. Continuing, QF participants were significantly more likely to respond correctly (i.e., choose 'different') than both the HF>QF participants and the HF participants for /ɛ ~ aɛ/ (mid contrast) (*OR* = 5.10, *z*-value = 2.91, *p* = 0.0416 and *OR* = 9.603, *z*-value = 5.7, *p* = 0.001, respectively) (Figure 6). In addition, as can be noted in Figure 6, HF>QF participants performed least accurately on this configuration. Finally, the model results also showed that QF participants performed better on the /ɛ ~ aɛ/ than on the /a ~ ɑ/ contrast (*OR* = 8.73, *z*-value = 3.8, *p* = 0.001).

**Table 12.** Post hoc table for the interaction between group and contrast for configuration 4–6.

| Contrast | Odds Ratio | SE | df | Null | z-Ratio | p-Value |
|---|---|---|---|---|---|---|
| HF low/HF>QF low | 0.50 | 0.27 | Inf | 1 | −1.30 | 0.783 |
| HF low/QF low | 0.53 | 0.31 | Inf | 1 | −1.10 | 0.882 |
| HF low/HF mid | 0.91 | 0.40 | Inf | 1 | −0.22 | 1.000 |
| HF low/HF>QF mid | 0.48 | 0.26 | Inf | 1 | −1.36 | 0.749 |
| HF low/QF mid | 4.64 | 2.58 | Inf | 1 | 2.76 | 0.064 |
| HF>QF low/QF low | 1.07 | 0.65 | Inf | 1 | 0.10 | 1.000 |
| HF>QF low/HF mid | 1.83 | 0.98 | Inf | 1 | 1.12 | 0.873 |
| HF>QF low/HF>QF mid | 0.97 | 0.50 | Inf | 1 | −0.06 | 1.000 |
| HF>QF low/QF mid | 9.31 | 5.53 | Inf | 1 | 3.76 | 0.002 |
| QF low/HF mid | 1.71 | 0.99 | Inf | 1 | 0.93 | 0.939 |
| QF low/HF>QF mid | 0.91 | 0.56 | Inf | 1 | −0.15 | 1.000 |
| QF low/QF mid | 8.74 | 4.94 | Inf | 1 | 3.84 | 0.002 |
| HF mid/MF mid | 0.53 | 0.29 | Inf | 1 | −1.18 | 0.846 |
| HF mid/QF mid | 5.10 | 2.85 | Inf | 1 | 2.91 | 0.042 |
| HF>QF mid/QF mid | 9.60 | 5.70 | Inf | 1 | 3.81 | 0.002 |

*4.5. Summary of Results*

With regards to our research question, i.e., what effect does extended exposure to QF as a D2 have on HF>QF listeners' ability to discriminate between the /a ~ ɑ/ and /ɛ ~ aɛ/ contrasts in QF, inferential analyses revealed no statistically significant differences in performance between the HF>QF participants and the HF participants for both contrasts and all configurations tested. If anything, what we found was that the HF>QF participants performed the least accurately of all three groups, i.e., that D2 acquisition of Quebec French seemed to impair their perception on this task rather than improve it. With regard to configurations 1–9 and 2–8, HF and HF>QF participants performed similarly to QF participants; however, not surprisingly, QF participants performed significantly better than both HF and HF>QF participants for configuration 3–7. Interestingly, with regard to configuration 4–6, QF participants performed better than both HF and HF>QF participants on the /ɛ ~ aɛ/ mid contrast but not on the /a ~ ɑ/ low contrast.

**5. Discussion**

The main aim of this study was to investigate the effects of extended exposure to QF as a D2 on the perceptual discrimination of the vowel contrasts /a ~ ɑ/ and /ɛ ~ aɛ/ in a group of native HF speakers who had moved to Quebec. A same–different discrimination task was conducted to determine whether increased exposure to non-native phonemic contrasts in a D2 can improve listeners' ability to perceive these as separate phonemes, as previously shown by, e.g., Bowie (2000). We hypothesized that the HF>QF group, having received extended exposure to QF, would be better at discriminating between subtly different pairs of words distinguished by the two target contrasts, as compared to the HF group, who had little first-hand experience with QF. If supported, this result would indicate that perceptual change had taken place among the mobile listeners after relocating to Quebec.

However, the results did not provide support for this hypothesis, as there was no evidence of the listeners in the HF>QF group showing a perceptual advantage over the HF group. Accordingly, it appeared that acquiring a new dialect did not enhance the perceptual capacity to perceive phonemic contrasts in that new dialect. Indeed, listeners in the HF>QF group proved overall to be, if anything, *less* accurate in their discrimination of differently paired /a ~ ɑ/ and /ɛ ~ aɛ/ items as compared to HF listeners. This pattern was revealed through descriptive differences between these two groups, which were particularly evident

for the 3–7 step configurations. Considered at face value, these results would seem to suggest that more exposure to a D2 can lead to *less* perceptual sensitivity of phonemic contrasts in the D2 and, therefore, that less exposure to D2 contrasts might in fact yield more perceptual sensitivity in discriminating these non-native phonemic contrasts.

It is interesting to speculate as to why a descriptively observable difference was found between the HF and HF>QF groups (with the latter performing less accurately), as this finding would appear to go against past studies showing that more exposure to D2 speech leads to *improved* abilities to perceive speech in this dialect (e.g., Scott and Cutler 1984; Bowie 2000; Walker 2018). One explanation for our surprising result could be that the QF variants /ɑ/ and /aɛ/ (i.e., phonemes largely absent from HF) have over time become less acoustically marked for the HF>QF participants and have potentially been integrated into the D1 phonemic categories /a/ and /ɛ/ as their exposure to D2 input has increased over time. This could also be related to the amount of attention that listeners in the HF>QF and HF groups pay to these contrasts. As Trudgill (1986, p. 11) argues, '[s]peakers are…more aware of variables whose variants are phonetically radically different' from corresponding variants in their native dialect. Thus, one could imagine that the HF>QF participants, who, compared to the HF participants, have had more acoustic input to QF as a D2, might be *less* aware of the variant phonemes /ɑ/ and /aɛ/ because of this D2 input. Therefore, the HF>QF listeners appear to be less able to discriminate between these phoneme categories and their native /ɛ/ and /a/ categories when the differences between contrasts /a ~ ɑ/ and /ɛ ~ aɛ/ are acoustically subtle (see also Auer et al. 1998).

However, given the fact that the inferential statistics failed to show an effect of mobility on group performance, i.e., no statistically significant difference was found between the HF and HF>QF groups in terms of their discrimination of any step configuration, at the very least, what we can say is that extended exposure to Quebec French did not appear to enhance the perceptual capacity to perceive phonemic contrasts in Quebec French. A lack of significance between HF and HF>QF listeners may well have been due to individual variation, particularly in the HF>QF group, as the amount of QF exposure among listeners in this group differed dramatically (ranging from 3 to 42 years). Due to space constraints, in the present study, we did not investigate intragroup variation among HF>QF participants (although, see Appendix C for information about individual participant performance). Future research should consider, e.g., correlational analyses employing nuanced measures of mobile participants' exposure to QF (e.g., personal social networks and media exposure in D2 country, see Voeten 2021; Ziliak 2012).

A second finding from this study was that participants in the QF group showed some evidence of an own-dialect advantage over the other two participant groups in their perception of the target contrasts, supporting results of past cross-dialectal perceptual studies (e.g., Adank et al. 2009; Clopper and Bradlow 2008; Dufour et al. 2007; Floccia et al. 2006; Impe et al. 2008). Namely, compared to the other two groups, QF listeners showed higher accuracy in discriminating the word pairs distinguished by the smallest acoustic differences (i.e., configurations 3–7 and 4–6). This advantage was particularly evident for QF listeners' discrimination of the /ɛ ~ aɛ/ contrast, as participants in this group showed high accuracy in discriminating between word pairs testing this contrast even at the most acoustically similar configurations of 3–7 and 4–6. One reason why QF listeners might have shown higher overall accuracy in discriminating between /ɛ ~ aɛ/ vs. /a ~ ɑ/ may be that the mid contrast shows more dynamic spectral movement across the length of the vowel (see Figure 3) as compared to the low contrast. Furthermore, the fact that the QF participants proved more similar to the other two groups in discriminating /a ~ ɑ/ vs. /ɛ ~ aɛ/ may be attributable to the fact that the former contrast is a 'phonetic creation' of QF (Reinke and Ostiguy 2016, p. 55), whereas the latter contrast has historically been present in HF. The different perceptual patterning found for each contrast could also be due to the different social evaluation of /a ~ ɑ/ vs. /ɛ ~ aɛ/ (with the low contrast being more socially neutral than the mid contrast in QF). This finding may also have to do with the fact that /a ~ ɑ/ is still undergoing merging in HF (Berns 2019), and thus is likely more

familiar than the /ɛ ~ aɛ/ contrast to HF speakers. For instance, Chalier (2021) showed that the opposition in /a ~ ɑ/ is still perceived among HF speakers, even if they do not make this contrast in production.

Future research on this topic could address some of the limitations of the current study. For instance, given that the conclusions of this study are based on relatively small sample sizex, replication studies with more participants could help determine whether the patterns identified in this study generalize to larger populations. Moreover, the potential effect of the relative unnaturalness of the experimental stimuli on the results should be explored in future work. In particular, given that the mid-vowel stimuli (i.e., *laide-l'aide*, *mettre-maître*) were rated as less natural-sounding than the low-vowel stimuli (i.e., *patte-pâte*, *tache-tâche*), future experiments would do well to normalize this aspect of the stimuli, to be able to mitigate any potential effects of perceived naturalness on perceptual discrimination. Notwithstanding these limitations, this study corroborates past research showing dialectal background to influence perceptual discrimination of vowel categories in French (e.g., Riverin-Coutlée and Arnaud 2015), thus advancing an understanding of the effects of native language background on speech perception more generally.

## 6. Conclusions

This study contributes to a growing body of research exploring the role that perceptual adaptation plays in SDA (see, e.g., Bowie 2000; Evans and Iverson 2004, 2007; Voeten 2021; Walker 2018; Ziliak 2012). Indirectly, it also contributes to studies in L1 perceptual attrition as it explores the malleability of the native language system upon new system acquisition (either the D2 or the L2). Surprisingly (see, e.g., de Leeuw et al. 2023; who show a perceptual advantage for bilingual returnees), we found some numerical evidence that HF participants with extended exposure to QF as a D2 showed *less* accuracy in discriminating between phonemic QF contrasts /a ~ ɑ/ and /ɛ ~ aɛ/ as compared to non-mobile HF speakers with little first-hand exposure to QF. Our finding that the HF>QF group was at least descriptively *less* accurate in their perception of non-native QF contrasts compared to the non-mobile HF group motivates further examination of both bilingual and bidialectal populations. Future studies on perceptual adaptation in SDA and SLA could help contribute to a unified theory of which perceptual capacities and variables are most likely to be influenced following D2/L2 contact.

**Author Contributions:** The research idea was conceptualized by S.K. with feedback from E.d.L. The experiment was designed and programmed by S.K. with feedback from E.d.L. Data were collected online by S.K. The statistics in R were conducted by E.P. with feedback from S.K. and E.d.L. The manuscript was mainly written by S.K. with E.P. writing the statistics/results sections and E.d.L. and E.P. giving feedback on all sections. The research was funded by a PhD scholarship to S.K. with E.d.L. as the primary supervisor. All authors have read and agreed to the published version of the manuscript.

**Funding:** This research was funded by the LISS DTP Research Training and Support Grant, made possible by a ESRC-funded PhD scholarship to S.K. with E.d.L. and Professor Leigh Oakes as supervisors.

**Institutional Review Board Statement:** The study was conducted in accordance with the Declaration of Helsinki and approved by the Institutional Review Board (or Ethics Committee) of Queen Mary University of London (QMERC20.559, 20th January 2022).

**Informed Consent Statement:** Informed consent was obtained from all subjects involved in the study.

**Data Availability Statement:** The present dataset can be found at the following link: https://drive.google.com/drive/folders/1fLlyHjgYJXcMdq8t2HcwaJM-PMPyu6kP?usp=sharing (accessed on 5 July 2023).

**Acknowledgments:** We would like to thank Leigh Oakes for his crucial advice when selecting the variables to test. We would also like to thank the audiences at Georgetown University Roundtable (GURT) 2022 and the Conference for the Association of French Language Studies 2022 for their useful feedback on earlier versions of this study. Finally, we would like to thank Marc Barnard, Cailtin Hogan, and Célia Richy for their helpful comments on earlier drafts of this article and the three external reviewers whose comments greatly improved this manuscript.

**Conflicts of Interest:** The authors declare no conflict of interest.

## Appendix A

**Table A1.** Word Frequency of Items Used in Same–Different Task (Accessed from the *Lexique 2* Corpus Using the CLEARPOND Database, Marian et al. 2012).

| Word Containing /a/ or /ɛ/ | Freq (per million) | Word Containing /ɑ/ or /aɛ/ | Freq (per million) |
|---|---|---|---|
| patte | 6.6 | pâte | 7.1 |
| tache | 13.7 | tâche | 26.4 |
| l'aide * | 294.4 | laide | 6.4 |
| mettre | 275.5 | maître | 121.8 |
| boule | 19.8 | bulle | 3.3 |
| épais | 6.25 | épée | 29.6 |
| bouche | 92.4 | bûche | 2.2 |
| saute | 36.7 | sotte | 4.75 |

* Note that the frequency reported for this word is for *aide* without the preceding article.

## Appendix B

**Table A2.** Duration and formant values for each step of each target continua. For each step, F1 and F2 measures are shown at three time points: at 25% of the vowel length, at 50%, and at 75%. Note that the uneven intervals between the formant values of continua steps are due to a Bark interpolation being used in the resynthesis process.

| Word | Step | Duration (s) | F1.25 | F1.50 | F1.75 | F2.25 | F2.50 | F2.75 |
|---|---|---|---|---|---|---|---|---|
| *patte-pâte* | 1 (*patte*) | 0.09558 | 1039.63 | 957.78 | 805.79 | 1845.46 | 1837.30 | 1771.33 |
| | 2 | 0.10246 | 1024.26 | 916.64 | 794.02 | 1826.20 | 1811.22 | 1708.24 |
| | 3 | 0.10245 | 968.70 | 883.64 | 789.80 | 1724.64 | 1672.78 | 1660.58 |
| | 4 | 0.11298 | 881.42 | 846.96 | 777.47 | 1616.57 | 1592.05 | 1545.00 |
| | 5 | 0.12308 | 850.95 | 851.60 | 759.80 | 1585.20 | 1582.62 | 1428.85 |
| | 6 | 0.13253 | 829.74 | 818.42 | 730.04 | 1576.10 | 1555.93 | 1402.02 |
| | 7 | 0.13655 | 821.45 | 825.70 | 730.79 | 1571.37 | 1452.24 | 1332.57 |
| | 8 | 0.14373 | 809.61 | 827.60 | 732.46 | 1525.39 | 1378.55 | 1176.40 |
| | 9 (*pâte*) | 0.13397 | 789.86 | 819.63 | 705.61 | 1406.95 | 1337.12 | 1176.50 |
| *tache-tâche* | 1 (*tache*) | 0.12551 | 880.52 | 895.70 | 780.15 | 1768.29 | 1695.66 | 1784.16 |
| | 2 | 0.13561 | 851.57 | 780.48 | 788.88 | 1694.10 | 1665.03 | 1691.33 |
| | 3 | 0.14623 | 822.93 | 730.28 | 778.66 | 1681.07 | 1462.57 | 1595.39 |
| | 4 | 0.15685 | 803.09 | 755.16 | 754.73 | 1635.42 | 1480.52 | 1494.33 |
| | 5 | 0.16502 | 768.18 | 741.37 | 727.58 | 1539.66 | 1466.01 | 1343.34 |
| | 6 | 0.17564 | 750.63 | 735.27 | 712.80 | 1466.85 | 1457.68 | 1284.72 |
| | 7 | 0.18544 | 742.25 | 729.05 | 677.04 | 1449.90 | 1451.57 | 1191.65 |
| | 8 | 0.19443 | 742.32 | 722.82 | 667.02 | 1444.62 | 1423.69 | 1094.64 |
| | 9 (*tâche*) | 0.20999 | 730.71 | 719.24 | 529.25 | 1430.76 | 1342.75 | 1030.46 |
| *mettre-maître* | 1 (*mettre*) | 0.07892 | 692.07 | 695.19 | 715.33 | 2332.98 | 2300.49 | 2194.68 |
| | 2 | 0.08332 | 707.07 | 707.25 | 718.82 | 2172.12 | 2149.95 | 2174.60 |
| | 3 | 0.09313 | 709.11 | 713.09 | 720.45 | 2057.60 | 2053.99 | 2082.72 |
| | 4 | 0.09634 | 718.18 | 718.77 | 717.53 | 1926.15 | 1940.09 | 1943.03 |
| | 5 | 0.10048 | 760.27 | 735.90 | 721.59 | 1883.08 | 1884.67 | 1917.19 |
| | 6 | 0.10293 | 762.13 | 748.06 | 727.65 | 1779.54 | 1788.87 | 1889.37 |
| | 7 | 0.10702 | 737.17 | 781.24 | 740.51 | 1733.20 | 1670.64 | 1783.27 |
| | 8 | 0.11355 | 749.57 | 831.36 | 762.64 | 1524.72 | 1560.47 | 1693.45 |
| | 9 (*maître*) | 0.15358 | 769.08 | 803.97 | 741.97 | 1490.80 | 1452.07 | 1515.82 |
| *l'aide-laide* | 1 (*laide*) | 0.15685 | 591.50 | 640.45 | 616.60 | 2380.82 | 2389.46 | 2343.92 |
| | 2 | 0.15685 | 658.25 | 671.14 | 632.43 | 2288.29 | 2294.31 | 2348.23 |
| | 3 | 0.16175 | 668.67 | 648.99 | 633.21 | 2258.70 | 2287.15 | 2276.90 |
| | 4 | 0.17155 | 674.74 | 664.48 | 636.97 | 2055.65 | 2182.91 | 2199.62 |
| | 5 | 0.17646 | 684.27 | 673.70 | 636.33 | 1865.51 | 2077.01 | 2089.26 |
| | 6 | 0.17482 | 744.58 | 719.75 | 636.10 | 1827.69 | 2021.39 | 2076.57 |
| | 7 | 0.1813 | 824.45 | 749.10 | 633.55 | 1801.30 | 1866.24 | 1959.17 |
| | 8 | 0.20750 | 859.87 | 718.60 | 631.62 | 1639.37 | 1847.95 | 1861.16 |
| | 9 (*l'aide*) | 0.21322 | 895.28 | 747.94 | 634.45 | 1620.58 | 1840.29 | 1840.90 |

## Appendix C

**Table A3.** Percentage of correctly perceived 'different' discriminations for the /a ~ ɑ/ (low) contrast by configuration (i.e., 1–9, 2–8, 3–7, 4–6) for each of the HF>QF participants. Note that each participant heard 4 different trials per configuration, where % correct showing 33% is because one of the trials within the given configuration timed out.

| Participant | Configuration | Contrast | % Correct |
| --- | --- | --- | --- |
| 3 | 19 | /a ~ ɑ/ | 75% |
| 3 | 28 | /a ~ ɑ/ | 75% |
| 3 | 37 | /a ~ ɑ/ | 25% |
| 3 | 46 | /a ~ ɑ/ | 25% |
| 4 | 19 | /a ~ ɑ/ | 100% |
| 4 | 28 | /a ~ ɑ/ | 100% |
| 4 | 37 | /a ~ ɑ/ | 25% |
| 4 | 46 | /a ~ ɑ/ | 50% |
| 7 | 19 | /a ~ ɑ/ | 100% |
| 7 | 28 | /a ~ ɑ/ | 100% |
| 7 | 37 | /a ~ ɑ/ | 100% |
| 7 | 46 | /a ~ ɑ/ | 25% |
| 8 | 19 | /a ~ ɑ/ | 100% |
| 8 | 28 | /a ~ ɑ/ | 75% |
| 8 | 37 | /a ~ ɑ/ | 25% |
| 9 | 19 | /a ~ ɑ/ | 100% |
| 9 | 28 | /a ~ ɑ/ | 100% |
| 10 | 19 | /a ~ ɑ/ | 100% |
| 10 | 28 | /a ~ ɑ/ | 100% |
| 10 | 37 | /a ~ ɑ/ | 100% |
| 15 | 19 | /a ~ ɑ/ | 100% |
| 15 | 28 | /a ~ ɑ/ | 75% |
| 15 | 37 | /a ~ ɑ/ | 75% |
| 15 | 46 | /a ~ ɑ/ | 50% |
| 16 | 19 | /a ~ ɑ/ | 100% |
| 16 | 28 | /a ~ ɑ/ | 100% |
| 16 | 37 | /a ~ ɑ/ | 100% |
| 16 | 46 | /a ~ ɑ/ | 25% |
| 17 | 19 | /a ~ ɑ/ | 100% |
| 17 | 28 | /a ~ ɑ/ | 75% |
| 17 | 37 | /a ~ ɑ/ | 75% |
| 17 | 46 | /a ~ ɑ/ | 25% |
| 18 | 19 | /a ~ ɑ/ | 100% |
| 18 | 28 | /a ~ ɑ/ | 100% |
| 18 | 37 | /a ~ ɑ/ | 75% |

**Table A3.** *Cont.*

| Participant | Configuration | Contrast | % Correct |
| --- | --- | --- | --- |
| 23 | 19 | /a ~ ɑ/ | 75% |
| 23 | 28 | /a ~ ɑ/ | 75% |
| 23 | 46 | /a ~ ɑ/ | 25% |
| 24 | 19 | /a ~ ɑ/ | 100% |
| 24 | 28 | /a ~ ɑ/ | 100% |
| 24 | 37 | /a ~ ɑ/ | 75% |
| 24 | 46 | /a ~ ɑ/ | 33% |

**Table A4.** Percentage of correctly perceived 'different' discriminations for the /ɛ ~ aɛ/ (mid) contrast by configurations (i.e., 1–9, 2–8, 3–7, 4–6) for each of the HF>QF participants. Note that each participant heard 4 different trials per configuration, where % correct showing 33% or 67% is because one of the trials within the given configuration timed out.

| Participant | Configuration | Contrast | % Correct |
| --- | --- | --- | --- |
| 3 | 19 | /ɛ ~ aɛ/ | 25% |
| 3 | 28 | /ɛ ~ aɛ/ | 100% |
| 3 | 37 | /ɛ ~ aɛ/ | 100% |
| 3 | 46 | /ɛ ~ aɛ/ | 50% |
| 4 | 19 | /ɛ ~ aɛ/ | 75% |
| 4 | 28 | /ɛ ~ aɛ/ | 75% |
| 4 | 37 | /ɛ ~ aɛ/ | 50% |
| 7 | 19 | /ɛ ~ aɛ/ | 75% |
| 7 | 28 | /ɛ ~ aɛ/ | 100% |
| 7 | 37 | /ɛ ~ aɛ/ | 75% |
| 8 | 19 | /ɛ ~ aɛ/ | 33% |
| 8 | 28 | /ɛ ~ aɛ/ | 50% |
| 9 | 19 | /ɛ ~ aɛ/ | 50% |
| 9 | 28 | /ɛ ~ aɛ/ | 50% |
| 9 | 46 | /ɛ ~ aɛ/ | 25% |
| 10 | 19 | /ɛ ~ aɛ/ | 100% |
| 10 | 28 | /ɛ ~ aɛ/ | 50% |
| 10 | 37 | /ɛ ~ aɛ/ | 25% |
| 15 | 19 | /ɛ ~ aɛ/ | 25% |
| 15 | 28 | /ɛ ~ aɛ/ | 75% |
| 15 | 37 | /ɛ ~ aɛ/ | 33% |
| 15 | 46 | /ɛ ~ aɛ/ | 25% |
| 16 | 19 | /ɛ ~ aɛ/ | 75% |
| 16 | 28 | /ɛ ~ aɛ/ | 100% |
| 16 | 37 | /ɛ ~ aɛ/ | 75% |

**Table A4.** *Cont.*

| Participant | Configuration | Contrast | % Correct |
|:---:|:---:|:---:|:---:|
| 17 | 19 | /ɛ ~ aɛ/ | 100% |
| 17 | 28 | /ɛ ~ aɛ/ | 100% |
| 17 | 37 | /ɛ ~ aɛ/ | 100% |
| 17 | 46 | /ɛ ~ aɛ/ | 50% |
| 18 | 19 | /ɛ ~ aɛ/ | 75% |
| 18 | 28 | /ɛ ~ aɛ/ | 100% |
| 18 | 37 | /ɛ ~ aɛ/ | 75% |
| 18 | 46 | /ɛ ~ aɛ/ | 50% |

## Notes

[1] A distinction between short-vowel [ɛ] and long-vowel [ɛ:] (as in the words [bɛl] *belle* 'beauty' and [bɛ:l] *bêle* 'bleat') was at one time ubiquitous in HF but has since been largely neutralized to the short variant. This merger of [ɛ] and [ɛ:] is similar to that between /a/ and /ɑ/ in the sense that it is nearing completion in most dialects of HF; however, it is different in that it is much further along in its neutralization compared to the /a ~ ɑ/ contrast. Gess (2008) argues that the vowel length distinction between [ɛ] and [ɛ:] was lost in HF by the sixteenth century. Nonetheless, vestiges of a length distinction between [ɛ] and [ɛ:] have been reported in HF, again mostly along the mid and upper eastern border of France (Avanzi 2017).

[2] The production tasks involved a word- and sentence-reading task involving the same five contrasts (two target and three distractor contrasts) discussed in this paper. All participants completed the production task first followed by the perception experiments in an attempt to hold constant any effects of the production task on the later perception experiments.

[3] Pairing the tokens in increasingly similar steps towards the *center* step in the vowel continua (e.g., steps 2–8, steps 3–7, 4–6) may not necessarily be the best way to test perceptual discrimination of these contrasts; in future development of this study, it would be worth testing whether pairing tokens of increasing similarity towards the *merged* category (e.g., steps 1–9, steps 1–7, steps 1–5, etc.) makes a difference in participants' discrimination of vowel continua steps.

[4] D' score measures indicate individuals' sensitivity to differences between stimuli and are designed to mitigate the influence of response biases (Macmillan and Creelman 2021), with a higher d' score indicating more sensitivity to a difference and a lower d' score indicating less sensitivity.

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
