# Peer review of "Perceptual Discrimination of Phonemic Contrasts in Quebec French: Exposure to Quebec French Does Not Improve Perception in Hexagonal French Native Speakers Living in Quebec"

_languages, doi:10.3390/languages8030193_

Round 1

Reviewer 1 Report

Review is attached.

Reviewer 2 Report

This is a very well-designed study. It contributes to the growing literature on D2 perception. An important contribution is in expanding the empirical scope of such research as most studies to date have focused on varieties of English. The focus on French vowel pairs that differ from each other in two different varieties is a good choice in terms of variable selection. Another important contribution is in showing that D2 exposure does not necessarily guarantee successful perceptual acquisition of new phonemic categories, though there are still some subtle differences between QF and mobile HF speakers living in Quebec that the author(s) address.

In terms of improvement, I have the following minor suggestions:

1) I would consider renaming the "bidialectal" group to the "mobile" group or something else. It seems that the focus of this study is to address whether or not this group is "bidialectal" in the first place (or at least in terms of acquiring perception of vowel contrasts in a D2) so calling them "bidialectal" seems to presume that they already are when in fact that is part of the question. I see that the authors actually do use the term "mobile" in Figures 7 and 8. So being consistent in the terms used to refer to each group would be helpful. But in general, I don't think bidialectal would be an appropriate term without also having production data and information about whether or not these participants see QF and HF as distinct dialects.

2) In lines 11-12, it says that "results showed some evidence". I suggest revising the abstract to be more specific about what that evidence is and what the results were. Were the results the same for both vowel pairs? The content in the last two or three sentences can be condensed to make room to present the specific results (if abstract length becomes a problem).

Reviewer 3 Report

The article reads well and the authors draw ok conclusions. I feel they could have discussed terms such as salience and markedness in more detail: it's clear that there are some, but small differences between the listener groups and the differences are really only clear for one of the token pairs. There could also be more information about the design such as whether there were pauses between blocks of tokens. There are some more comments and also some suggestions for changes in the attached file.

Round 2

Reviewer 1 Report

Review is attached.

Round 3
